# Predicting Stroke Etiology with Radiomics: A Retrospective Study

**DOI:** 10.3390/medsci13030098

**Published:** 2025-07-26

**Authors:** Jacobo Porto-Álvarez, Antonio Jesús Mosqueira Martínez, Javier Martínez Fernández, José L. Taboada Arcos, Miguel Blanco Ulla, José M. Pumar, María Santamaría, Emilio Rodríguez Castro, Ramón Iglesias Rey, Pablo Hervella, Pedro Vieites Pérez, Manuel Taboada Muñiz, Roberto García-Figueiras, Miguel Souto Bayarri

**Affiliations:** 1Department of Radiology, Hospital Clínico Universitario de Santiago de Compostela, 15706 Santiago de Compostela, Spain; jacobo.porto.alvarez2@sergas.es (J.P.-Á.); miguel.souto@usc.es (M.S.B.); 2Neurorradiología, Health Research Institute of Santiago de Compostela (IDIS), 15706 Santiago de Compostela, Spain; 3Stroke Unit, Department of Neurology, Hospital Clínico Universitario, 15706 Santiago de Compostela, Spain; 4Neuroimaging and Biotechnology Laboratory (NOBEL), Clinical Neurosciences Research Laboratory (LINC), Health Research Institute of Santiago de Compostela (IDIS), 15706 Santiago de Compostela, Spain; 5Department of Dermatology, Hospital Clínico Universitario de Santiago de Compostela, 15706 Santiago de Compostela, Spain; 6Department of Anaesthesia, Hospital Clínico Universitario de Santiago de Compostela, 15706 Santiago de Compostela, Spain

**Keywords:** acute ischemic stroke, AIS, stroke etiology, clot, thrombus composition, radiomics, machine learning, artificial intelligence

## Abstract

**Background/Objectives**: The composition of the thrombus is not taken into account in the etiology determination of patients with acute ischemic stroke (AIS); however, it varies depending on the origin of the thrombus, as atherothrombotic thrombi contain more red blood cells and cardioembolic thrombi contain more fibrin and platelets. Radiomics has the potential to provide quantitative imaging data that may vary depending on the composition of thrombi. The aim of this study is to predict cardioembolic and atherothrombotic thrombi using radiomic features (RFs) from non-contrast computed tomography (NCCT) brain scans. **Methods**: A total of 845 RFs were extracted from each of the 41 patients included in the study. A predictive model was used to classify patients as either cardioembolic or atherothrombotic, and the results were compared with the TOAST criteria-based classification. **Results**: Ten RFs (one shape feature and nine texture features) were found to demonstrate a statistically significant correlation with cardioembolic or atherothrombotic origins. The predictive radiomics model achieved an area under the curve (AUC) of 0.842 and an accuracy of 0.902 (*p* < 0.001) in classifying stroke etiology. **Conclusions**: Radiomics based on NCCT can help to determine the etiology of AIS.

## 1. Introduction

Every year, 15 million people worldwide suffer a stroke, resulting in 5 million deaths and 5 million individuals with significant disabilities for the remainder of their lives [1]. In 2021, it was estimated that 795,000 people in the United States would experience a stroke, with 85% of cases being ischemic [2]. The underlying causes of acute ischemic stroke (AIS) are not always easily identified. Nevertheless, the identification of its etiology is crucial for the management of patients, given that it is a leading cause of morbidity and mortality on a global scale. The etiology of AIS is diagnosed based on a combination of analytical and clinical parameters, cardiological testing, and parameters derived from qualitative analysis of radiological images. The TOAST (Trial of Org 10172 in Acute Stroke Treatment) criteria were developed to determine the origin of thrombi causing AIS and divide their probable etiology into five different groups: cardioembolic, atherothrombotic, lacunar infarct, unusual origin, and indetermined origin [3]. Whilst the diagnosis of a lacunar or unusual stroke is relatively straightforward, the classification of an AIS as either a cardioembolic or atherothrombotic event is not always straightforward and is of significant clinical importance. In the context of secondary prevention management, patients with an atheromatous etiology will typically be administered antiplatelet therapy, whereas those with a cardioembolic origin will generally be provided with anticoagulant therapy. In many cases, patients may exhibit features associated with both etiological groups, resulting in an unknown etiology, or an “indetermined” classification according to the TOAST criteria. This hinders the formulation of effective secondary prevention strategies for these patients.

Furthermore, the molecular composition of a thrombus varies depending on the underlying cause [4,5]. Thrombi of atherothrombotic origin contain a greater proportion of red blood cells in comparison to those resulting from other etiologies classified within the TOAST system. In contrast, thrombi of cardiogenic origin exhibit a higher proportion of fibrin and platelets compared to those caused by other factors [6,7,8]. This difference in molecular composition may therefore be a reason for a potentially different radiological behavior of these two types of thrombi. There have been reports of radiological differences between cardioembolic and atherothrombotic thrombi. The thrombi of cardioembolic origin have been shown to exhibit greater density and attenuation on Non-Contrast Computed Tomography (NCCT) [9,10]. This finding indicates that the radiological manifestation of the thrombus is contingent on its molecular composition and, consequently, its origin. Nevertheless, despite the molecular and radiological differences observed, no molecular or radiological criterion related to clots is utilized in the classification of the etiology of acute AIS.

Radiomics is a field of radiology that focuses on the extraction and analysis of a large number of quantitative data from radiological images that correlate with the underlying pathophysiology [11]. Numerous studies have been conducted on this tool, with a particular focus on oncological pathology. These studies have demonstrated a correlation between radiomic data and various molecular patterns, genetic mutations, and other biological phenomena. However, recent studies using this technique outside oncological settings, particularly for neurologic diseases, have become increasingly common [12]. Given the established differences in the molecular composition and radiological appearance of thrombi between etiologies of AIS, radiomics may provide further insight into these differences by analyzing quantitative data from images of thrombi. There are already a few recently published articles showing that radiomic data can provide important information to determine the different origins of thrombi in patients with AIS [13,14].

The hypothesis of this study is that the different molecular composition of thrombi of atherothrombotic and cardioembolic origin will result in a disparate radiomic pattern of thrombi from these two etiology groups on brain NCCT. This may provide valuable information in determining the etiology of AIS. Therefore, the objective of this article is to employ a machine learning model based on radiomic data obtained from thrombi in NCCT scans of patients with AIS, for the purpose of classifying them as cardioembolic or atherothrombotic etiology.

## 2. Materials and Methods

### 2.1. Study Design

This prospective case–control study was conducted in accordance with the Declaration of Helsinki of the World Medical Association (2008) and approved by the local Ethics Committee of Santiago-Lugo (code 2023/299) [15]. The patients were selected from the database of patients with suspected AIS who were treated at University Hospital of Santiago de Compostela, a public third-level hospital, between 1 January 2021 and 31 December 2021 (with a total of 882 patients). Informed consent was obtained from each patient after a full explanation of the procedures. All patients received treatment from expert neurologists and neuroradiologists from the Clinical Hospital of Santiago de Compostela (Spain) in accordance with national and international guidelines.

### 2.2. Patients

The study’s inclusion criteria were limited to: (1) patients with AIS caused by thrombi in the internal carotid artery (ICA) and middle cerebral artery (MCA) (M1 and proximal M2 segments); (2) patients with NCCT performed using a slice thickness of less than 1 mm; (3) patients with visible clot on NCCT; and (4) follow-up visits three months after a stroke in living patients. The study’s exclusion criteria were as follows: (1) patients with AIS who had undergone NCCT and computed tomography angiography (CTA) in a different hospital; (2) patients with AIS resulting from other procedures, such as aneurysmal or tumor embolization; (3) patients with more than one occluded intracranial vessel or tandem occlusion; (4) patients with etiology other than cardioembolic or atherothrombotic according to TOAST criteria; and (5) patients with dual cardioembolic and atherothrombotic etiology according to TOAST criteria; (6) patients suspected of having cardioembolic or atherothrombotic etiology but do not meet the main criteria defined by the TOAST system for each group; and (7) patients with occlusion of the distal middle cerebral artery (M3 or M4 segments).

### 2.3. Image Acquisition

All patients enrolled in the study underwent an NCCT at our hospital utilizing one of two different CT scanners (16 rows of detectors, 120 kV) of the same make and model (Phillips Ingenuity; Amsterdam, The Netherlands) during the diagnosis process of AIS. Patients were randomly assigned to each scanner. The images obtained had a slice thickness of 0.625 mm. Although reconstructions with a thickness of 1 mm were available, they were not used for analysis. The window width and center were set at 80 and 40 Hounsfield units, respectively (Figure 1).

### 2.4. Segmentation, Preprocessing, and Feature Extraction

Two interventional neuroradiologists and a radiology resident who had undergone specialized training performed semi-automated segmentation of each thrombus. The segmentation was conducted using the open-access software 3D Slicer (version 5.2.2, Massachusetts, USA) [16]. The software includes a segmentation tool (Level Tracing tool) that enables semi-automatic segmentation based on automatic edge detection. The region of interest segmented was the clot visible on NCCT in patients with AIS (Figure 2). Segmentation was performed in the axial, sagittal, and coronal planes (Figure 3). The window width and center were set to 100 and 50 HU, respectively.

Radiomic features were obtained using the Slicer Radiomics tool, which is also available in 3D-Slicer [17]. This application uses the computational classes implemented in the Pyradiomics library. During the feature extraction process, 3D-Slicer allows image voxel resampling and kernel size modification. These parameters were not modified. Conversely, the images were normalized by smoothing with a Gaussian filter and a fixed value of 25 for the gray bin width, and wavelet-based features were also extracted. The complete set of features available in 3D Slicer was extracted, encompassing the following: first order, GLCM, GLDM, GLRLM, GLSZM, NGTDM, and shape-based features. A total of 32,110 RF were obtained, with 845 RF for each patient included in the study.

The radiomics quality score (RQS) was developed to measure the quality of radiomic studies [18]. Our study received a score of 19 out of 36 (52.78%) (Appendix B). Furthermore, the preparation of this article adheres to the CheckList for EvaluAtion of Radiomics research (CLEAR) guidelines [19] (see Appendix A).

The segmentation, extraction of RF, and analysis of the results were performed using a system with an Intel CORE i7 processor (Intel Corporation, Santa Clara, Santa Clara Country, CA, USA), 16 GB RAM, 1 TB hard disk, and Microsoft Windows 11 operating system (Microsoft Corporation, Redmond, King Country, WA, USA).

### 2.5. Clinical Data

The study also recorded the median Hounsfield units (HU) of the clot for each etiological group. Other clinical data were also recorded, including age, sex, the presence of hypertension, diabetes mellitus, dyslipidemia, alcohol and drug use, and smoking. In addition to these data, information is available on the tPA administration, the laterality of the thrombus, the ASPECTS score, and the degree of collaterality following the ASITN/SIR collateral grading scale [20]. The treatment of patients at the time of stroke was not a consideration in the analysis. The patient’s condition is measured before and after treatment using the modified Ranking Scale (mRS) and the NIHSS scale.

### 2.6. Statistical Analysis

The RF selection and the analysis of RF and clinical variables were conducted using Statistical Package for the Social Sciences Statistics (SPSS) (version 21, IBM Armonk, New York, NY, USA) [21]. Firstly, a multivariate analysis was conducted, employing a logistic regression model to ascertain the variables associated with the two etiologies of AIS, with a 95% confidence interval. The multivariate analysis incorporated 845 RFs and 9 clinical variables (age; sex; arterial hypertension; drug, alcohol, or smoke consumption; diabetes; dyslipidemia; and Hounsfield units). In order to select significant variables, the *p*-value must fall below 0.05. With regard to the remaining clinical data, the administration of tPA was not considered due to its occurrence subsequent to the NCCT procedure, thereby rendering the radiomic data antecedent to this administration. The ASPECTS score and the patient’s functional status were not considered in the analysis due to the fact that the focus of the segmentation is exclusively on the thrombus, excluding the brain parenchyma.

The predictive models were constructed with the open-access software Orange: Data Mining Toolbox in Python (version 3.33.0, Ljubljana, Slovenia) [22]. A total of three predictive models were constructed, namely: (i) a Radiomics model, based on the RF that emerged as the most statistically significant according to the multivariate analysis; (ii) a clinical model, comprising solely clinical variables; and (iii) a combined model that incorporated both the selected RF and the clinical variables (Figure 4). The automatic classifier utilized was a Neural Network, a multi-layer perception algorithm also available from Orange Data Mining [23,24]. The Orange software suite facilitates the modification of parameters associated with the Neural Networks classifier. The configuration parameters for the classifier are as follows: 100 neurons per hidden layer, the ReLu activation function for the hidden layer, a stochastic gradient-based optimizer (Adam) for weight optimization, and 200 maximal iterations.

The Orange application employed for the evaluation of the performance of the classification model is “Test and Score”. Test and Score permits the implementation of diverse sampling methodologies. In this instance, the sampling method that was employed was leave-one-out cross-validation (LOOCV). The LOOCV method selects *n* − 1 patients for the training group, with the remaining patients being allocated to the test group. This process is repeated *n* times, with a different patient being assigned to the test group on each occasion. The LOOCV method is particularly recommended for evaluating the performance of machine learning models when the number of datasets is limited [25]. Test and Score also permits the observation of the classifier performance measures. The classification accuracy and area under the curve (AUC) of the predictive models were calculated with this application. In addition to the aforementioned functionality, the application facilitates the integration of supplementary widgets, including “confusion matrix” widget, which serves to provide a visual representation of the confusion matrix of the classifiers, and “box plot” widget, which quantifies the concordance between the classifier results and the actual classification by employing a chi-square test and a 95% confidence interval.

The classifier’s performance in the three models is measured using the Cohen’s kappa coefficient (K), the AUC, the accuracy, the sensitivity (Se), and the specificity (Sp), with a 95% confidence interval. The Kappa coefficient is a statistical measure of the extent to which the true and predicted categories are aligned, excluding the possibility of agreement by chance. Its value is more conservative and statistically more valid than the balanced accuracy or AUC. The confusion matrix is a graphical representation of the relationship between the predictions made by Neural Networks (represented by the columns of the matrix) and the TOAST criteria-based classification (represented by the rows of the matrix) (Table 1). The true positive (*TP*) is defined as the number of atherothrombotic AIS patients correctly identified as such. The false positive (*FP*) is defined as the number of atherothrombotic AIS patients incorrectly identified as cardioembolic AIS patients. The true negative (*TN*) is defined as the number of cardioembolic AIS patients correctly identified as such. Finally, the false negative (*FN*) is defined as the number of cardioembolic AIS patients incorrectly identified as atherothrombotic AIS patients.

The Kappa coefficient (in %) is defined for classification problems with two categories, in our case, Atherothrombotic AIS and Cardioembolic AIS, as(1)K=100(Pa−Pe)/(1−Pe)
were $Pa=\;\left(TP+TN\right)/N$ and $Pe=\left(TP+FN\right)\left(TP+FP\right)/N^{2}+\left(FP+TN\right)\left(FN+TN\right)/N^{2}$. The *Se* is defined as the classifier’s ability to correctly detect patients with atherothrombotic AIS, while the *Sp* is defined as the classifier’s ability to correctly detect patients with cardioembolic AIS. The *Se*, *Sp*, and Accuracy are defined by:(2)Se=TP/(TP+FN)(3)Sp=TN/(TN+FP)(4)$$Accuracy=\frac{\left(TP+TN\right)}{\left(TP+FP+TN+FN\right)}$$

## 3. Results

### 3.1. Patient Selection

Out of 882 patients, only 41 were selected based on the inclusion and exclusion criteria. These patients were divided into two groups using the TOAST system: cardioembolic (29 patients) and atherothrombotic (12 patients) etiology (Figure 5 and Table 2).

### 3.2. Feature Reduction

In the multivariate analysis performed with SPSS (version 21), of the 845 RFs extracted, only 10 were statistically significantly associated with cardioembolic and atherothrombotic etiology of AIS (*p*-value < 0.05) (Table 3). The features that were selected for inclusion in the study included 1 shape feature (Sphericity) and 9 texture features: 4 Gray-Level Dependence Matrix (GLDM), 2 Gray-Level Co-occurrence Matrix (GLCM), 2 Gray-Level Run Length Matrix (GLRLM), and 1 Neighborhood Gray Tone Difference Matrix (NGTDM). The shape features describe morphological aspects of the region of interest. The GLDM features are responsible for determining the dependency of voxels in a given neighborhood on a single center voxel. The GLCM features calculate the frequency with which adjacent pixels of each gray-level value co-occur. The GRLM features are metrics that quantify the number of lines of a specific gray level and the length that occur in a given direction. Finally, the NGTDM features are metrics that analyze the difference between the gray value of a pixel and that of its immediate vicinity [26].

Of the clinical variables included in the multivariate analysis, none were shown to have a statistically significant association with the cardioembolic and atherothrombotic etiology of AIS (*p*-value > 0.05) (Table 4).

### 3.3. Prediction Models

The radiomic model demonstrated the capacity to differentiate between the two types of thrombi and accurately predict the patients’ cardioembolic and atherothrombotic etiology of AIS. The accuracy, AUC, Se, and Sp for predicting stroke etiology were 0.902, 0.842, 0.833, and 0.931, respectively (*p*-value 0.000), with Kappa = 76.43% (Table 5 and Figure 6).

However, when the RF and the clinical variables (combined model) were employed, the accuracy, AUC, Se, and Sp for predicting stroke etiology decreased to 0.732, 0.655, 0.556, and 0.781, respectively (*p*-value 0.040), with a Kappa = 30.07% (Table 6) (Figure 6).

The clinical model showed the worst performance in predicting the etiology of AIS, with statistically non-significant results, with an accuracy of 0.561, an AUC of 0.402, a Se of 0.300, and a Sp of 0.710 (*p*-value 0.993), with a Kappa = −6.03% (Table 7 and Figure 6).

## 4. Discussion

The present study has demonstrated the capacity of radiomics to differentiate between cardioembolic and atherothrombotic thrombi. The molecular differences between these two types of clots also reflect a difference in imaging representation, thus establishing a correlation between the RF of NCCT images and the atherothrombotic and cardioembolic etiology of AIS. A total of 845 RFs were analyzed; however, only a subset of 10 RFs that were statistically associated with these two etiological groups (*p* < 0.05) were selected for further investigation. Multivariate analysis revealed no statistically significant association between these two etiologies of AIS and the clinical variables investigated, including clot density, arterial hypertension, dyslipidemia, diabetes mellitus, smoking, alcoholism, drug use, age, and sex (*p*-value > 0.05). Three predictive models were developed: one based on RF alone, one based on clinical variables alone, and a third model based on the combination of RF with clinical variables. An automatic classifier based on neural networks (Neural Network) has been used. The radiomic model performed very well, with an AUC of 0.842, an accuracy of 0.902, a Se of 0.833, and a Sp of 0.931. The model’s performance, as measured by Cohen’s Kappa index (K = 76.43%), demonstrated substantial agreement with the TOAST criteria, which are recognized as the gold standard for the etiological classification of AIS. However, when clinical variables were introduced into the model, its predictive performance was found to deteriorate, with the clinical model demonstrating the most unfavorable outcomes.

The present findings are consistent with those reported in two other articles published on the subject of the prediction of the etiology of AIS. Chen et al. obtained an AUC of 0.9018 and an accuracy of 0.8929 in differentiating between cardioembolic and atherothrombotic etiology using radiomic features based on CTA images [13]. The most notable difference between the two studies is the source of the radiomic data. In the present work, the radiomic data are obtained from the NCCT, while in the referenced article, they are obtained from the CTA. A further distinction between our work and the referenced article is that we perform a semi-automatic segmentation, while they employed a manual segmentation. The semi-automated segmentation performed is based on automatic edge detection, with the radiologist responsible for ensuring that the segmentation includes as much of the thrombus area as possible. In patients with an arterial clot visible on NCCT, the contrast between the region of interest and the rest of the brain parenchyma is sufficiently remarkable to be easily detected by the automatic edge detection method, with the radiologist only intervening to accept or correct the segmentation performed. This made the segmentation faster and included the entire thrombi. Finally, the aforementioned article does not incorporate clinical variables within the radiomic analysis, in contrast to the approach employed in the present article.

Regarding the other published article, Jiang J et al. obtained an AUC of 0.838 in predicting the cardioembolic etiology of AIS in a sample of 403 patients, also using manual segmentation. They used NCCT-based radiomic features of patients with AIS [14]. As far as this article is concerned, the main difference lies in the fact that in our case, we are trying to predict both etiological atherothrombotic and cardioembolic groups, instead of limiting ourselves to predicting only one of them. The segmentation process is also manual, as described by Chen et al. Furthermore, this article makes no mention of clinical variables in the context of radiomic analysis. On the other hand, the images used in this case are also from NCCT, which also gives good results in predicting the cardioembolic group, supporting our findings that there is a correlation between the radiomic data obtained from NCCT and the etiology of thromboembolic events in patients with AIS. Therefore, this article also concluded that radiomics could be helpful in determining the etiology of AIS.

Determining the etiology of AIS is crucial for effective therapeutic management and early implementation of appropriate secondary prevention measures [27]. The classification of a stroke as lacunar or of infrequent etiology using the TOAST (Trial of Org 10172 in Acute Stroke Treatment) criteria is well-protocolized. However, in cases of cardioembolic and atherothrombotic etiology, the boundaries may be less clearly defined, resulting in a significant number of patients being labeled as having an “undetermined etiology”. In other cases, the information for etiology determination is only available after the acute onset of stroke, leading to delayed identification of the cause of AIS. The intention of this study is to utilize radiomics in order to provide additional information that will assist in the classification of patients who meet the criteria for both etiological groups, or whose etiology has been incompletely studied (classified as “indetermined” according to the TOAST criteria). However, thrombi of atherothrombotic and cardioembolic origin exhibit divergent molecular compositions [5,6,7,8], yet these specific molecular data remain inaccessible in the acute care setting for these patients. Conversely, radiomic data derived from NCCT are obtainable early in the management of patients with AIS. The present study makes a significant contribution to the extant literature by demonstrating that radiomics also has the capacity to differentiate thrombi of atherothrombotic origin from those of cardioembolic origin. These findings may assist in the timely and accurate diagnosis of the etiology of stroke in such patients.

Regarding the limitations of our study, the first one is that it is a retrospective study. In this regard, since there is not much literature available, we believe that the first step to investigate whether radiomics can contribute something to the diagnosis of the etiology of AIS is to perform a retrospective study, as it is the one that involves the least ethical conflicts, as well as not delaying or altering the usual management of these patients. Having shown that the association appears to exist with a retrospective study, we believe that the next step is to confirm these findings with a prospective study. Another classic limitation of radiomic studies is external validity. In our case, images from two different CT scanners of the same make and model were used. In this sense, it is necessary to include images from scanners of different manufacturers and from other hospitals to increase the external validity of these studies. For this reason, we believe that multicenter studies are also needed, because single-center studies seem to show that such an association exists. Finally, another limitation of radiomic studies is the difference in methodology between study groups in data processing and analysis of radiomic variables. In this case, it is necessary to publish in detail the steps carried out in order to increase the available bibliography in this field and to share methodologies that can be reproduced by other research groups, with the aim of unifying the analytical processes as much as possible. In terms of specific limitations of our study, it is important to note that we had a lower number of subjects in comparison to previous studies. This may limit the applicability of the study to clinical practice. Furthermore, the study exclusively includes patients with visible thrombus on NCCT, thereby limiting the generalizability of the results to those patients in whom the thrombus is not visible on NCCT. In our case, in addition to a significantly shorter recruitment period, the fact that only patients with a clot visible on NCCT and pure occlusion of the distal ICA or proximal branches of the MCA were selected meant that the N was not higher. With this in mind, a sampling method recommended for low N studies was used (LOOCV). Further patient recruitment is needed to increase the sample size and to include other patient groups not analyzed in the current article. The incorporation of additional imaging techniques and biomarkers may also result in an increase in the number of patients [28]. With regard to the segmentation process, no study of interobserver variability has been conducted. Instead, the segmentations have been reviewed by a group of neuroradiologists who are experts in diagnosis and interventional procedures. Lastly, it is important to note that patient medication and thrombus age have not been considered in the present study, nor in any previously published research. These factors may vary in both etiological groups. In this regard, the relationship between antiplatelet and anticoagulant therapy, which could have the capacity to modify the composition of thrombus, has not been evaluated. This limitation has also been identified in the previously published studies, and it is recommended that it be explored in future research.

## 5. Conclusions

Radiomic features can help classify patients with AIS into cardioembolic or atherothrombotic etiology, with consequent benefit in patient management. The present article confirms the hypothesis that molecular differences between thrombi of cardioembolic and atherothrombotic origin also translate into radiomic differences between these two etiology groups. This provides significant data that may facilitate the classification of the etiology of AIS.

## Figures and Tables

**Figure 1 medsci-13-00098-f001:**
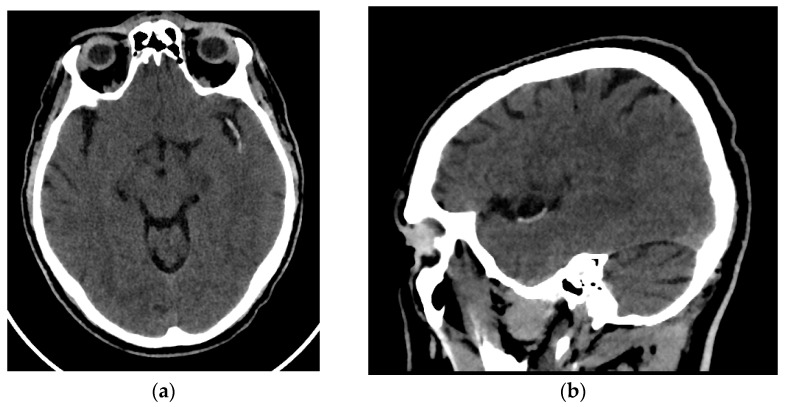
Brain NCCT of a patient with AIS and the hyperdense MCA sign. This is one of the radiological signs of AIS in NCCT. (**a**) Axial NCCT scan of a patient with a hyperdense left MCA sign. (**b**) Sagittal NCCT scan of the same patient.

**Figure 2 medsci-13-00098-f002:**
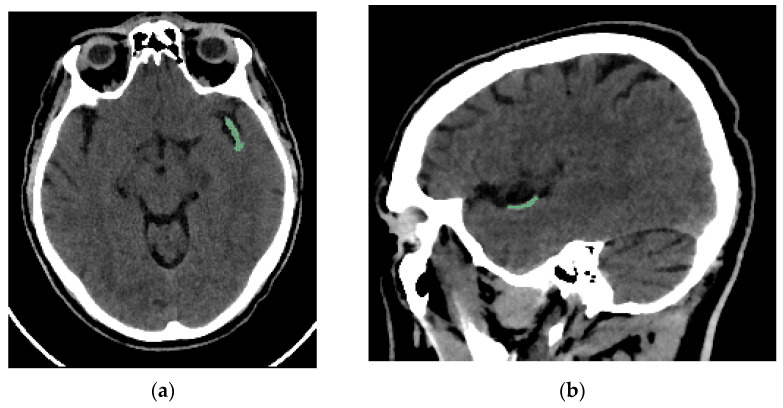
Brain NCCT of the same patient as in Figure 1, with the thrombus segmented. The segmentation was performed using the “Level Tracing” tool of 3D Slicer. (**a**) Axial NCCT with thrombus segmented. (**b**) Sagittal NCCT of the same patient with the thrombus segmented.

**Figure 3 medsci-13-00098-f003:**
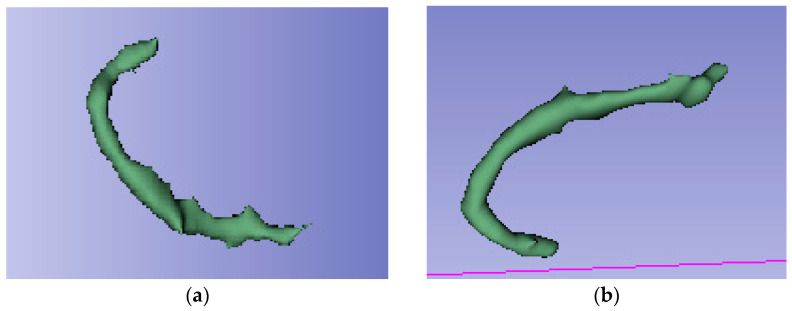
3D reconstruction of the segmented thrombus of the same patient as in Figure 1 and Figure 2. Segmentation is performed in all 3 spatial planes with 3D Slicer. (**a**) Oblique coronal view of the segmented thrombus. (**b**) Segmented thrombus seen in oblique caudal view.

**Figure 4 medsci-13-00098-f004:**
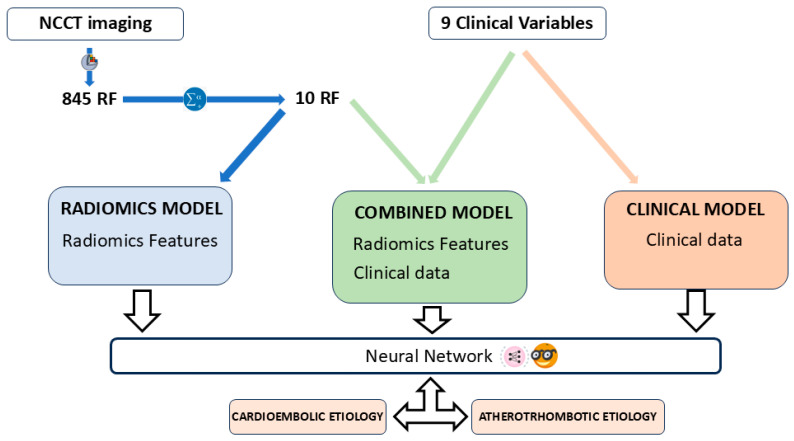
Article workflow. Three prediction models were developed: a radiomics model with the selected RF, a combined model with RFs and clinical data, and a clinical model with clinical data only. The automatic classifier used was a Neural Network, available at Orange: Data Mining Toolbox in Python.

**Figure 5 medsci-13-00098-f005:**
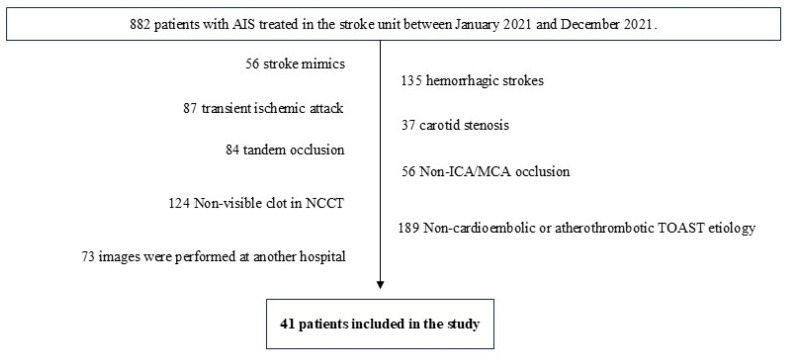
Following the implementation of the inclusion and exclusion criteria, a total of 41 patients were selected for inclusion in the study.

**Figure 6 medsci-13-00098-f006:**
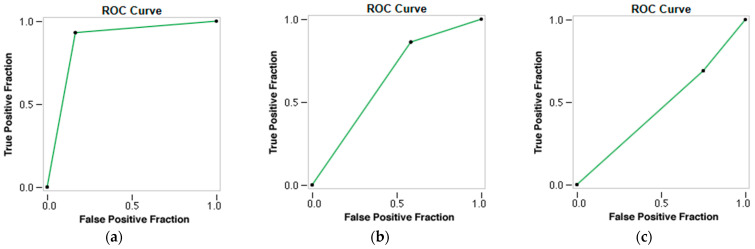
ROC curves of the three prediction models using a Neural Network classifier. (**a**) ROC curve of the Radiomics Model. (**b**) ROC curve of the Combined Model. (**c**) ROC curve of the Clinical Model.

**Table 1 medsci-13-00098-t001:** Representation of a confusion matrix used to visualize the performance of a neural network classifier. Columns represent the predicted class. The rows represent the true class according to the TOAST criteria.

		Predicted with Neural Network	
		Atherothrombotic	Cardioembolic
TOAST	Atherothrombotic	TP	FP
	Cardioembolic	FN	TN

**Table 2 medsci-13-00098-t002:** After the inclusion and exclusion criteria, 41 patients were included.

41 Patients Included	
Cardioembolic etiology	29 (70.73%)
Atherothrombotic etiology	12 (29.26%)
Female sex	22 (53.66%)
Age (mean)	72.90 (SD 12.56)
Arterial hypertension	29 (70.73%)
Diabetes mellitus	13 (31.71%)
Dyslipidemia	23 (56.09%)
Smoking	7 (17.07%)
Alcohol	6 (14.63%)
Drug	1 (2.44%)
Hounsfield units (mean)	62.73 (SD 11.83)
Clot on right ICA	4 (9.76%)
Clot on right MCA	17 (41.46%)
Clot on left ICA	3 (7.31%)
Clot on left MCA	17 (41.46%)
ASPECTS (mean)	8.58 (SD 1.22)
Collateral score system < 2	5 (12.20%)
mRS previous (mean)	1.08 (SD 1.17)
mRS at 3 months (mean)	3.18 (SD 1.84)
NIHSS initial (mean)	15.16 (SD 4.59)
NIHSS at 24h (mean)	7.87 (SD 6.96)

**Table 3 medsci-13-00098-t003:** RF that showed a statistically significant association with the etiology of AIS in the multivariate analysis performed in SPSS, using the logistic regression method.

Radiomics Features	Coeff. *	RF Class	OR	*p*-Value
Sphericity	6.797	Shape	8.952 × 10^5^	0.049
Imc1 (2)	18.526	GLCM	1.11135 × 10^19^	0.039
Cluster Tendency (4)	33.426	GLCM	3.286 × 10^14^	0.036
Large Dependence Low Gray-Level Emphasis (4)	0.072	GLDM	1.074	0.015
Large Dependence Low Gray-Level Emphasis (6)	0.060	GLDM	1.062	0.027
Long Run Low Gray-Level Emphasis (6)	2.252	GLRLM	9.508	0.037
Dependence Variance (7)	0.409	GLDM	1.505	0.017
Short Run Low Gray-Level Emphasis (7)	−28.260	GLRLM	6.331 × 10^−13^	0.041
Complexity (7)	−48.639	NGTDM	1.000 × 10^−13^	0.045
Dependence Variance (8)	0.492	GLDM	1.636	0.045

* Coeff. = Coefficient.

**Table 4 medsci-13-00098-t004:** Clinical variables included in the multivariate analysis did not show a statistically significant relationship with the etiology of AIS.

Clinical Features	Cardioembolic (29)	Atherothrombotic (12)	Coeff.	OR	*p*-Value
Female sex	16 (55.17%)	6 (50%)	−0.208	0.813	0.763
Age (mean)	74.55 (SD 13.26)	68.91 (SD 10.07)	0.037	1.037	0.193
Arterial hypertension	22 (75.86%)	7 (58.33%)	−0.809	0.445	0.267
Diabetes mellitus	8 (27.59%)	5 (17.24%)	0.629	1.875	0.381
Dyslipidemia	15 (51.72%)	8 (66.67%)	0.624	1.867	0.384
Smoking	4 (13.79%)	3 (25%)	0.606	1.833	0.481
Alcohol	3 (10.34%)	3 (25%)	1.099	3.000	0.229
Drug	1 (3.45%)	0 (0%)	−20.356	1.444 × 10^−9^	1.000
Hounsfield units (mean)	63.14 (SD 13.15)	61.75 (SD 8.21)	0.010	1.011	0.730

**Table 5 medsci-13-00098-t005:** Confusion matrix of the radiomics model (utilizing only RF), with the automatic classifier Neural Network.

		Neural Network (Radiomics Model)		∑
		Atherothrombotic	Cardioembolic	
**TOAST**	**Atherothrombotic**	10	2	**12**
	**Cardioembolic**	2	27	**29**
**∑**		**12**	**29**	

**Table 6 medsci-13-00098-t006:** The following confusion matrix illustrates the performance of the combined model (utilizing RF and clinical variables) with the automatic classifier Neural Network.

		Neural Network (Combined Model)		∑
		Atherothrombotic	Cardioembolic	
**TOAST**	**Atherothrombotic**	5	7	**12**
	**Cardioembolic**	4	25	**29**
**∑**		**9**	**32**	

**Table 7 medsci-13-00098-t007:** Confusion matrix of the clinical model (utilizing only clinical variables), also with the automatic classifier Neural Network.

		Predicted with Neural Network		∑
		Atherothrombotic	Cardioembolic	
**TOAST**	**Atherothrombotic**	3	9	**12**
	**Cardioembolic**	9	20	**29**
**∑**		**12**	**29**	

## Data Availability

The radiomic analysis software used for the prediction models is open access (Orange Data Mining). The code of the automatic classifier (neural network) used is available at https://scikit-learn.org/stable/modules/neural_networks_supervised.html (accessed on 20 July 2025). Patient images and radiomic data are not published for ethical reasons.

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
