# Peer review of "Predicting Stroke Etiology with Radiomics: A Retrospective Study"

_medsci, 2025, doi:10.3390/medsci13030098_

Round 1

Reviewer 1 Report

Comments and Suggestions for Authors

This is a report on a single-centre prospective study of the use of radiomic factors (RF) to determine if an intravascular clot seen on non-contrast CT brain scan of an acute stroke patient was cardioembolic or atherothrombotic in origin. A total of 845 RFs were extracted from each of the 41 out of 882 patients included in the study. A predictive model was used to classify patients as either 27 cardioembolic or atherothrombotic, and the results were compared with the TOAST criteria-based classification. The authors identified 10 RFs that were significantly correlated, with a high accuracy. These results may be helpful clinically.

However, there are a number of issues the authors may wish to address:

  1. Line 20 - what is ‘defemination’?
  2. Line 20 – the comma (,) in ‘(AIS), however’ may be better replaced with a semi-colon (;)
  3. Line 29 - the 10 RFs can be listed here
  4. Line 30-21 – ‘The clinical features included in the study did not show a statistically significant correlation with thrombus etiology’ may be unfair – it was never claimed to be able to do so, and is thus best removed
  5. Line 33 - ‘p=0.000’ – do the authors mean ‘p<0.001’?
  6. Lines 66-67 - There have been reports of radiological differences between cardioembolic and atherothrombotic thrombi’ – some of these ‘differences’ can be listed here with appropriate references
  7. Section 2.2 – the clinical features collected need to be listed and defined, although they are mentioned in lines 178-179. Was medication being taken a time of stroke considered?
  8. Line 125 - ‘using two’ - this gives the impression that the patients were scanned on both machines – ‘using’ may be better replaced with ‘in one of’
  9. Line 138 – clarify the ‘three spatial planes’
  10. Line 166 and throughout the paper – use ‘sex’ in place of ‘gender’, unless it was gender that was looked at
  11. Table 4 – the ‘n’ should be given at the top with ‘cardioembolic’ and ‘atherothrombotic’, and % be added to the numbers for categorical data
  12. (major) Line 247 – a flow diagram showing reasons (with numbers) for exclusion is needed
  13. Lines 341-364 – this limitations para should come just before ‘Conclusions’. That only 41 of 882 patients could be studied may show the limited value of this study to clinical practice
  14. (major) The Discussion should also cover the impact of the age of the clot, effects of anti-thrombotics the patient had been taking at time of stroke, and clot size, on the radiomics

Author Response

The authors would like to express their gratitude to the reviewer for the insightful questions 
and contributions. Should any further modifications be required, we will be at the reviewer's 
disposal. 

Commentary 1: Line 20 - what is ‘defemination’?

There was a grammatical error (determination). Corrected. Thank you (line 22).

Commentary 2: Line 20 – the comma (,) in ‘(A2IS), however’ may be better replaced with a semi-colon (;)

Checked, thank you (line 22).

Commentary 3: Line 29 - the 10 RFs can be listed here

Due to the word limit of the abstract, it is not possible to list all RFs. Instead, we indicate that one shape feature and nine texture features were extracted. Thank you very much (line 31).

Commentary 4: Line 30-21 – ‘The clinical features included in the study did not show a statistically significant correlation with thrombus etiology’ may be unfair – it was never claimed to be able to do so, and is thus best removed

Deleted, thank you.

Commentary 5: Line 33 - ‘p=0.000’ – do the authors mean ‘p<0.001’?

Corrected (line 34).

Commentary 6: Lines 66-67 - There have been reports of radiological differences between cardioembolic and atherothrombotic thrombi’ – some of these ‘differences’ can be listed here with appropriate references

Along the same lines, it is specified immediately below that thrombi of cardioembolic origin showed greater density and attenuation in non-contrast brain CT scans, with references 9 and 10 attached. Thank you very much (lines 68-70). 

Commentary 7: Section 2.2 – the clinical features collected need to be listed and defined, although they are mentioned in lines 178-179. Was medication being taken a time of stroke considered?

Section 2.5 of the clinical data reflects the non-radiomic variables that were considered. We would like to add that the medication taken by patients at the time of their stroke was not considered. We would like to thank the reviewer for suggesting this, as it was not something we had considered when preparing the study (lines 171-172).

Commentary 8: Line 125 - ‘using two’ - this gives the impression that the patients were scanned on both machines – ‘using’ may be better replaced with ‘in one of’

Corrected. Thak you very much (lines 126-127).

Commentary 9: Line 138 – clarify the ‘three spatial planes’

The segmentation was performed on the three radiological planes (axial, sagittal, and coronal). The text has been corrected to facilitate understanding. I would like to express my sincere gratitude (lines 139-140).

Commentary 10: Line 166 and throughout the paper – use ‘sex’ in place of ‘gender’, unless it was gender that was looked at

Corrected. Thank you very much (lines 167 and 180).

Commentary 11: Table 4 – the ‘n’ should be given at the top with ‘cardioembolic’ and ‘atherothrombotic’, and % be added to the numbers for categorical data

A column marked 'N' has been incorporated into each of the clinical variables, accompanied by the relevant percentage (Table 4).

Commentary 12: (major) Line 247 – a flow diagram showing reasons (with numbers) for exclusion is needed

Included (Figure 5). Thak you very much.

Commentary 13: this limitations para should come just before ‘Conclusions’. That only 41 of 882 patients could be studied may show the limited value of this study to clinical practice

Corrected. The sentence has been added that says the low N limit the applicability of the study to clinical practice. Thank you very much (lines 389-390).

Commentary 14: (major) The Discussion should also cover the impact of the age of the clot, effects of anti-thrombotics the patient had been taking at time of stroke, and clot size, on the radiomics

It has been posited that the present study, as well as those that have been published previously, do not analyse patients' medication or the time taken for thrombi to develop. These are variables that should be analysed in future studies. Although the length of the thrombus has not been analysed, its morphology and volume have been examined in terms of its shape. We would like to express our gratitude to the reviewer for this valuable contribution (lines 331-334).

Reviewer 2 Report

Comments and Suggestions for Authors

This retrospective study investigates whether radiomic features extracted from non-contrast CT can differentiate cardioembolic from atherothrombotic thrombi in acute ischemic stroke (AIS). Using 845 radiomic features from 41 patients and a neural network classifier, the model showed high accuracy (AUC 0.842). Clinical variables alone had poor predictive power. The results support the utility of radiomics for early stroke etiology classification.

Major:

External Validation: The model was developed and tested within a single-center dataset using LOOCV. Have the authors considered validating their model with external datasets to assess generalizability?

Radiomic Feature Stability: Given the known sensitivity of radiomic features to acquisition parameters and segmentation variability, can the authors comment on inter-observer variability and feature robustness?

Clinical Integration: While clinical variables did not improve the model, how do the authors envision integrating radiomic predictions into real-time clinical workflows, especially in cases with “undetermined” TOAST classification?

Additional Imaging Biomarkers for Ischemic Regions: Recent studies have also highlighted the utility of other imaging biomarkers such as quantitative susceptibility mapping (QSM) and oxygen extraction fraction (OEF) mapping, other than radiomics. A brief comparison with these modalities would enhance the paper's depth and relevance. Please consider citing the following references:

10.1161/STROKEAHA.123.044606

Author Response

The authors would like to express their gratitude to the reviewer for the insightful questions and contributions. Should any further modifications be required, we will be at the reviewer's disposal.

Comment 1: External Validation: The model was developed and tested within a single-center dataset using LOOCV. Have the authors considered validating their model with external datasets to assess generalizability?

One of our future goals is to include an external cohort. This has not been possible in the current study due to limitations with the ethics committee, which only granted us permission to analyse patients from our hospital. We have requested permission to include patients from other hospitals and to conduct a prospective study. Thank you very much for this insightful question.

Comment 2: Radiomic Feature Stability: Given the known sensitivity of radiomic features to acquisition parameters and segmentation variability, can the authors comment on inter-observer variability and feature robustness?

One limitation of this study is that no interobserver variability study was conducted on segmentation. Instead, the process was overseen by a team of three experienced neuroradiologists specialising in neurodiagnostics and neurointervention. We have added a sentence to the discussion section to address this limitation ( lines 398-400).

Comment 3: Clinical Integration: While clinical variables did not improve the model, how do the authors envision integrating radiomic predictions into real-time clinical workflows, especially in cases with “undetermined” TOAST classification?

The authors believe that, in future, a rapid analysis tool for radiomic variables will be integrated into PACS. Similar to the tools currently available for measuring density in Hounsfield units, it would provide a probability of atherothrombotic or cardioembolic etiology. This could be used to modify the diagnostic tests performed on these patients in cases where the basic battery of tests does not reveal a clear etiology. For example, the period of cardiac monitoring could be extended to look for atrial fibrillation that did not show up in the basic monitoring performed on these patients.

Comment 4: Additional Imaging Biomarkers for Ischemic Regions: Recent studies have also highlighted the utility of other imaging biomarkers such as quantitative susceptibility mapping (QSM) and oxygen extraction fraction (OEF) mapping, other than radiomics. A brief comparison with these modalities would enhance the paper's depth and relevance. Please consider citing the following references: 10.1161/STROKEAHA.123.044606

Very interesting article (reference 27). It has been added to the text that it would be interesting to include new imaging tests and new imaging biomarkers (line 397).

Round 2

Reviewer 1 Report

Comments and Suggestions for Authors

This is a revised submission report on a single-centre prospective study of the use of radiomic factors (RF) to determine if an intravascular clot seen on non-contrast CT brain scan of an acute stroke patient was cardioembolic or atherothrombotic in origin.

I thank the authors for addressing most of my concerns, the paper is much improved.

However, a few remain:

  1. Line 34 – ‘p<0.000’ – do the authors mean ‘p<0.001’?
  2. Table 4 – I apologise if I was not clear earlier. My request is eg

1st row - Cardioembolic (n = 29), Atherothrombotic (n = 12)

2nd row – 16 (55.2%) ( = 15/49), 6 (50%) ( = 6/12) thus, as a % of the column, not row

  1. Line 331 – ‘patient medication’ - the specific mention of anti-platelets and anticoagulants at time of stroke occurrence is needed as these can affect clot composition
  2. Lines 331-334 – best placed in Limitations
  3. Lines 367-370 - best moved to become 1st 2 sentences of the Conclusion
  4. Lines 389-390 – My earlier comment that ‘only 41 of 882 patients could be studied may show the limited value of this study to clinical practice’ refers to the fact that only 41/882 = 4.6% had clots that were analysable. Many patients do not have visible clots, hence the limitation of the clinical applicability of this new knowledge

Author Response

We would like to express our gratitude to the reviewer for their contribution to the enhancement of this article.

Comment 1: Line 34 – ‘p<0.000’ – do the authors mean ‘p<0.001’?

Revised, thak you.

Comment 2: Table 4 – I apologise if I was not clear earlier. My request is eg: 1st row - Cardioembolic (n = 29), Atherothrombotic (n = 12); 2nd row – 16 (55.2%) ( = 15/49), 6 (50%) ( = 6/12) thus, as a % of the column, not row

We regret the misunderstanding. The error has been rectified. Thank you.

Cooment 3: Line 331 – ‘patient medication’ - the specific mention of anti-platelets and anticoagulants at time of stroke occurrence is needed as these can affect clot composition. 

Corrected. We mentioned in lines 397-400 that antiplatelet therapy and anticoagulant therapy have not been included in this study or in the referenced studies. It is acknowledged that this may modify the composition of the thrombus, and it is recommended that this be investigated in future studies.
We would like to reiterate our gratitude for this intriguing concept, which we intend to incorporate into our future research endeavours.

Comment 4: Lines 331-334 – best placed in Limitations

Corrected (lines 394-400). Thank you.

Comment 5: Lines 367-370 - best moved to become 1st 2 sentences of the Conclusion

Moved (lines 404-408). Thank you.

Comment 6: Lines 389-390 – My earlier comment that ‘only 41 of 882 patients could be studied may show the limited value of this study to clinical practice’ refers to the fact that only 41/882 = 4.6% had clots that were analysable. Many patients do not have visible clots, hence the limitation of the clinical applicability of this new knowledge

The 882 patients were treated in the stroke unit of the hospital. The study population included patients who did not have an ischemic stroke (haemorrhagic stroke, stroke mimics, transient ischaemic attack) and others who, despite having a stroke and visible thrombus, were not included because they could not be classified as cardioembolic or atherothrombotic according to the TOAST criteria. Consequently, 41 patients were identified, constituting 10.84% of the total sample of 378 patients (see Figure 5). It is hypothesised that the N could be increased if distal thrombi were included or if CT angiography were used to segment thrombi not visible on non-contrast brain CT. The inclusion of these patients was precluded by the fact that CT angiography only demonstrates the contrast stop, thus delineating the theoretical location of the thrombus. However, it is possible that a more distal occlusion or a thrombus of different size could stop the contrast. We therefore hypothesise that this would include areas that we would not know if they belonged to the thrombus. This is the underlying reason for the observed low number of patients. We added a comentary (lines 383-385). I would like to express my sincere gratitude for your insightful and precise evaluation.

Reviewer 2 Report

Comments and Suggestions for Authors

Authors addressed my concerns properly.

Author Response

Thank you very much.